# Metabolic Syndrome—Role of Dietary Fat Type and Quantity

**DOI:** 10.3390/nu11071438

**Published:** 2019-06-26

**Authors:** Peter Clifton

**Affiliations:** School of Pharmacy and Medical Sciences, University of South Australia, Adelaide SA 5000, Australia; peter.clifton@unisa.edu.au

**Keywords:** carbohydrate, polyunsaturated fat, monounsaturated fat, saturated fat, fish oil, meta-analyses, lipids, glucose, blood pressure, insulin resistance

## Abstract

Background: Metabolic syndrome increases the risk of cardiovascular disease (CVD) over and above that related to type 2 diabetes. The optimal diet for the treatment of metabolic syndrome is not clear. Materials and Methods: A review of dietary interventions in volunteers with metabolic syndrome as well as studies examining the impact of dietary fat on the separate components of metabolic syndrome was undertaken using only recent meta-analyses, if available. Results: Most of the data suggest that replacing carbohydrates with any fat, but particularly polyunsaturated fat, will lower triglyceride(TG), increase high density lipoprotein (HDL) cholesterol, and lower blood pressure, but have no effects on fasting glucose in normal volunteers or insulin sensitivity, as assessed by euglycemic hyperinsulinemic clamps. Fasting insulin may be lowered by fat. Monounsaturated fat (MUFA) is preferable to polyunsaturated fat (PUFA) for fasting insulin and glucose lowering. The addition of 3–4 g of N3 fats will lower TG and blood pressure (BP) and reduce the proportion of subjects with metabolic syndrome. Dairy fat (50% saturated fat) is also related to a lower incidence of metabolic syndrome in cohort studies.

## 1. Introduction

The metabolic syndrome is associated with an increased risk of cardiovascular disease and type 2 diabetes and enhances the risk of CVD in people with diabetes [1]. There are currently four definitions of metabolic syndrome, three of which require insulin resistance, as evidenced by central obesity, or fasting hyperinsulinemia or a disturbance in glucose homeostasis, plus two other abnormalities [2]. The National Cholesterol Education Program Adult Treatment Panel (NCEP ATP-III (revision 2005) guidelines does not require this, but demands 3 of 5 possible abnormalities—central obesity, impaired fasting glucose, hypertriglyceridemia, low HDL cholesterol, and blood pressure (>130 systolic or >85 diastolic). The other criteria vary with different levels of blood pressure (140/90), with a combination of high TG and low HDL as one criterion, and varying levels of fasting TG—either 1.7 or 2 mmol/L—and varying levels of HDL cholesterol. This review will focus on the components of the metabolic syndrome and where there are data on the presence or absence of metabolic syndrome, the criterion used will be mentioned. The aim of this review is to systematically examine meta-analyses that summarise the evidence from interventions on fat amount and type in metabolic syndrome or its separate components. Some evidence from individual trials that are illustrative will be included.

## 2. Aim

To systematically review meta-analyses of interventions that replace carbohydrates with fat in people with metabolic syndrome and interventions that examine these effects on the individual components of the syndrome. 

## 3. Methods

Pubmed was searched (all years available) with the terms “meta-analysis AND dietary fat AND carbohydrate AND intervention AND (TG, OR, HDL, OR blood pressure, OR glucose, OR weight)”. We reviewed 102 titles.

## 4. Results 

### 4.1. Lipids

A large amount of evidence has accumulated that replacing carbohydrates (usually quality unspecified) with fat of any sort will lower fasting triglyceride and increase HDL cholesterol. A meta-analysis by Mensink and Katan [3] found that a 1% increase in fat calories in place of carbohydrates led to a fall in TG of 0.026 mmol/L (95% confidence intervals: 0.020–0.031) with polyunsaturated fat, 0.021 (0.015–0.027) with saturated fat, and 0.019 (0.014–0.024) with monounsaturated fat from 100 studies with 45 diets. For HDL cholesterol the same diets led to an increase in HDL cholesterol of 0.006 (0.003–0.009) mmol/L, 0.010 (0.007–0.013), and 0.008 (0.005–0.011), respectively. Overall saturated fat is about 30% more effective than the other two fats in the combined lowering of TG and HDL cholesterol. However, differences in HDL cholesterol from genetic variance have not been associated with differences in coronary heart disease (CHD) risk [4], whereas TG lowering genetically has been associated with CVD reduction, with the same degree of benefit per mg/dL apoB lowering as a reduction in LDL cholesterol [5]. Thus, PUFA is preferred to other fat types for carbohydrate replacement. In a small study of 39 men with metabolic syndrome, PUFA produced greater TG lowering than MUFA (both 5–30% of energy). Overall, 25% (4 of 16) assigned to PUFA and 13% (3 of 23) to MUFA did not have metabolic syndrome after the intervention [6]. Metabolic syndrome in this study was based on NCEP-III (2001). A high total dairy intake (and, presumably, a lower carbohydrate diet) is associated with a 6% reduced risk of metabolic syndrome per additional serving of dairy [7]. Most studies in this meta-analysis of 16 case-control/cross-sectional studies used the NCEP-III criteria. 

The relationship between carbohydrate intake and TG has been controversial for many years with arguments about the persistence of the TG elevation effect [8,9], with the moderating or even nullifying effect of fibre on TG elevation [10], the contrasting effects of higher versus lower sugar with sugars replacing starch [11,12], and the absence of TG elevation in Pima Indians with type 2 diabetes mellitus (DM) with increased carbohydrates [13]. Just when the landscape was reasonably predictable from interventions, the TOSCA.IT showed in 18,785 people with type 2 DM that increasing fat intake from <25% to >35% increased TG, while increasing carbohydrate intake from <45% to >65% decreased TG [14]. TG was lower in the highest tertile of the relative Mediterranean diet score, which had lower added sugars, more fibre, but less fat and more carbohydrates. [15]. However, the TOSCA-IT is a large cohort study and its results may well be confounded as they do not match the results from dietary intervention studies in people with type 2 DM. The Qian meta-analysis [16] of 24 studies with 1460 participants showed that fasting TG was reduced by 0.31 mmol/L (95% confidence interval −0.44, −0.18) with replacement of carbohydrates with MUFA. A high fibre intake is associated with a 30% lowering in the risk of metabolic syndrome [17].

In a meta-analysis of high-fat versus low-fat diets in people with obesity, but no overt metabolic disturbance, Lu et al. (2018) [18] found a significantly higher level of TG (WMD: 11.68 mg/dL (0.13 mmol/l), 95 % CI 5.90, 17.45; *p <* 0.001) and a lower level of HDL-cholesterol (WMD: −2.57 mg/dL (−0.07 mmol/l); 95 % CI −3.85, −1.28; *p <* 0.001) after the low-fat diets, compared with high-fat diets in 20 studies with 2016 participants.

### 4.2. Fish Oil Fatty Acids

Guo et al. [19] performed a meta-analysis of seven case-control and 20 cross-sectional studies and found that a higher level of plasma/serum *n*-3 PUFAs was associated with a lower metabolic syndrome risk (pooled OR = 0.63, 95% CI: 0.49, 0.81). The plasma/serum *n*-3 PUFAs in controls were significantly higher than in metabolic syndrome cases (WMD: 0.24; 95% CI: 0.04, 0.43), especially docosapentaenoic acid and docosahexaenoic acid. 

The addition of fish oil fatty acids of at least 1 g/day lowers fasting TG and a metanalysis performed by Eslick et al. [20] of 47 studies showed that taking fish oils (weighted average daily intake of 3.25 g of EPA and/or DHA) produced a clinically significant reduction of TG (−0.34 mmol/L, 95% CI: −0.41 to −0.27), with a very slight increase in HDL (0.01 mmol/L, 95% CI: 0.00 to 0.02) and LDL cholesterol (0.06 mmol/L, 95% CI: 0.03 to 0.09). The reduction of TG correlated with EPA plus DHA intake and initial TG level.

### 4.3. Glucose

Fasting glucose lowering by reducing carbohydrate and replacing it with fat is far more controversial. A recent meta-analysis by Wanders et al. [21] showed no effect in normal subjects of replacing carbohydrate with polyunsaturated fat, even though fasting insulin was reduced. A 5% increase in energy from PUFA significantly reduced insulin by 5.8 pmol/L (95% CI −10.2 to −1.3  pmol/L), but not glucose (change −0.07, 95% CI −0.17 to 0.04  mmol/L) and even in the group with the highest intake of PUFA, glucose was still not significant (−0.09, 95% CI −0.18 to 0.01 mmol/L). Imamura et al. [22] found that replacing 5% energy from carbohydrates with SFA had no significant effect on fasting glucose (+0.02 mmol/L, 95% CI = −0.01, +0.04; *n* trials = 99), but lowered fasting insulin (−1.1 pmol/L; −1.7, −0.5; *n* = 90). Replacing saturated fat with PUFA lowered fasting glucose (0.04mmol/L; 0.01, 0.07). Thus, PUFA is clearly the better fat for replacing carbohydrates in normal people. In people with type 2 diabetes [16], high MUFA diets compared with high carbohydrate diets lowered fasting plasma glucose (WMD −0.57 mmol/L [95% CI −0.76, −0.39]) in 24 studies containing 1460 participants. HDL cholesterol was increased by 0.06 mmol/L (0.02, 0.10) [16]. Surprisingly, in this study, replacing PUFA with MUFA lowered fasting glucose by a large amount (−0.87 mmol/L (−1.67, −0.07)) but the data are much less reliable, taken from only four studies and 44 participants.

### 4.4. Blood Pressure

There are much less data on blood pressure and carbohydrate replacement with fat in non-diabetics and the effects are relatively small. A meta-analysis performed by Shah et al. [23] found that diets rich in carbohydrates resulted in significantly higher systolic blood pressure (difference: 2.6 (95% CI: 0.4, 4.7) mm Hg; *p* = 0.02) and diastolic blood pressure (1.8 (0.01, 3.6) mm Hg; *p* = 0.05) than did diets rich in cis-monounsaturated fat. Huntress et al. [24] examined low carbohydrate diets in people with type 2 diabetes from data at one year. Eighteen trials were included in the meta-analysis, which found that the low carbohydrate diet lowered systolic blood pressure (estimated effect  =  −2.74  mmHg, 95% CI −5.27 to −0.20), but diastolic BP was not significant. In the meta-analysis from Qian et al. [16], MUFA lowered systolic blood pressure (−2.31 mmHg (−4.13, −0.49)) when it replaced carbohydrates.

### 4.5. Diet Composition During Weight Loss and Lipid Changes

Mansoor et al. [25] performed a meta-analysis of 11 weight loss trials with 1369 participants with a low carbohydrate level being defined as <20% carbohydrates. Compared with participants on low fat diets, participants on low carbohydrate diets experienced a greater reduction in body weight (weighted mean difference [WMD] −2.17 kg; 95% CI −3.36, −0.99) and TG (WMD −0.26 mmol/L; 95% CI −0.37, −0.15), but a greater increase in HDL-cholesterol (WMD 0·14 mmol/l; 95% CI 0.09, 0.19) and LDL-cholesterol (WMD 0·16 mmol/L; 95% CI 0.003, 0.33). Most of the low carbohydrate diets followed an Atkins style diet with an increase in saturated fat, which accounted for the rise in LDL cholesterol. If carbohydrates are replaced by unsaturated fat and not saturated fat, no rise in LDL cholesterol is seen in either six months [26], or one- [27] or two-year follow ups [28].

### 4.6. Insulin Resistance

Insulin resistance is a key and essential element of the metabolic syndrome (except the NCEP111 criteria), usually assumed on the basis of central adiposity. As noted by Wanders et al. [21], replacing carbohydrates with PUFA led to a lowering of fasting insulin, as did saturated fat, suggesting reduced insulin resistance, at least in the liver. There have been a small number of formal hyperinsulemic euglycemic clamp studies, but no meta-analysis. Tardy et al. [29] compared high dairy and industrial trans fatty acids with low trans fat diets in 63 healthy women with abdominal obesity. After four weeks of 60 g low-TFA lipids/day (0.54 g/day; *n* = 21), ruminant TFA-rich lipids (4.86 g/day; *n* = 21), or industrial TFA-rich lipids (5.58 g/day; *n* = 21), no changes in peripheral insulin sensitivity were seen. Bendtsen et al. [30] found no effect either of 15 g/day of trans fat in partially hydrogenated soybean oil for 16 weeks in 52 overweight postmenopausal women. Fasching et al. [31] found no effect of exchanging 200 g of carbohydrates with 90 g of PUFA, MUFA, or saturated fat for one week in a randomised crossover study in eight men with insulin sensitivity, assessed with a euglycemic hyperinsulinemic clamp. Borkmann et al. [32] found no effect of substituting saturated fat for carbohydrates on insulin sensitivity in eight non-diabetic subjects, despite large changes in LDL and TG (the latter down 33%.) The KANWU study [33] showed that a high MUFA diet for three months reduced insulin resistance, compared with a high saturated fat diet in 162 healthy subjects, but carbohydrate levels were not examined. Fish oil had no effect and the effect of MUFA was lost when fat intake was >37% of energy. In a very small study in patients with fatty liver disease, a high MUFA Mediterranean diet improved insulin sensitivity, compared with a high carbohydrate diet (*p* = 0.03) accompanied by a reduction in liver fat [34]

A contrary result was found in the Lipgene study [35] where 472 volunteers with metabolic syndrome were randomised to one diet for 12 weeks: High MUFA or high saturated fat diets or high carbohydrate diets with and without fish oil (1.2 g/day). In the highest HOMA-IR tertile, MUFA and n3 fats lowered insulin significantly, compared with saturated fat. In the lowest HOMA-IR tertile, insulin and glucose rose with all diets, but it rose less with MUFA and N3 fats compared with saturated fat. There is regression to the mean in both these tertiles and there is no statistical contrast between the effect of diets in the different tertiles, so we don’t actually know if there is a tertile/diet interaction. Triglycerides fell with N3 fats in tertiles 1 and 2, but surprisingly, not in the highest tertile with the highest TG level. Replacing carbohydrates with MUFA or saturated fat had no effect on TG in any tertile, which is contrary to the much bigger meta-analysis of Mensink et al. (3) and there is no good explanation other than strong time-related changes.

In another report from the same study, in 337 volunteers [36], the prevalence of metabolic syndrome (NECP-III) fell by 20.5% after the n-3 diet (blood pressure and TG fell), compared with the high saturated fat diet (10.6%), high MUFA diet (12%) diet, and high carbohydrate diet (10.4%) (*p <* 0.028).

## 5. Conclusions

Most meta-analyses show that replacement of carbohydrates with fat lowers fasting TG and glucose and blood pressure, and increases HDL cholesterol with some differences, depending on whether the population has type 2 diabetes or not. There are some large intervention and cohort studies that show the opposite results, but these are in the minority. PUFA is probably superior to MUFA, while fish oil is superior to both.

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
