# Peer review of "Metabolic Syndrome—Role of Dietary Fat Type and Quantity"

_nutrients, 2019, doi:10.3390/nu11071438_

Reviewer 1 Report

The purpose of the study has not been clearly defined. The author has not described the literature selection strategy. Only brief information on this subject was given in the summary. Please specify from which years the meta-analyzes were included in the study.

It is not clear which definition of Metabolic syndrome was accepted by the author.

Moreover, the study is actually not about the metabolic syndrome itself, as the title suggests, but it is about its components.

The author also does not give any conclusion at the end.

Author Response

Response to reviewer 1

I Have changed the abstract, added an introduction, aims, methods and conclusions.

RE controversies. I have added a comment about TOSCA-IT as a cohort study and not an intervention and confounding may be in place. The Lipgene  study I think has time related changes and regression to the mean and division into HOMA tertiles has complicated it with no overall analysis. The bulk of the data does not support their results.

CHO/fat swap-no differences whether it is ketogenic or not. CHO/protein swap not examined as fat was the focus.

Abstract:  Background. Metabolic syndrome increases the risk of CVD over and above that related to type 2 diabetes. The optimal diet for the treatment of metabolic syndrome is not clear. Materials and Methods. A review of dietary interventions in volunteers with metabolic syndrome as well as studies examining the impact of dietary fat on the separate components of metabolic syndrome was undertaken using only recent meta-analyses if available. Results. Most of the data suggests that replacing carbohydrate with any fat, but particularly polyunsaturated fat, will lower TG, increase HDL cholesterol, lower blood pressure but have no effects on fasting glucose in normal volunteers or insulin sensitivity as assessed by euglycemic hyperinsulinemic clamps. Fasting insulin may be lowered by fat.  MUFA is preferable to PUFA for fasting insulin and glucose lowering. The addition of 3-4g of N3 fats will lower TG and BP and reduce the proportion of subjects with metabolic syndrome. Dairy fat (50% saturated fat) is also related to a lower incidence of metabolic syndrome in cohort studies.

Abstract:  Background. Metabolic syndrome increases the risk of CVD over and above that related to type 2 diabetes. The optimal diet for the treatment of metabolic syndrome is not clear. Materials and Methods. A review of dietary interventions in volunteers with metabolic syndrome as well as studies examining the impact of dietary fat on the separate components of metabolic syndrome was undertaken using only recent meta-analyses if available. Results. Most of the data suggests that replacing carbohydrate with any fat, but particularly polyunsaturated fat, will lower TG, increase HDL cholesterol, lower blood pressure but have no effects on fasting glucose in normal volunteers or insulin sensitivity as assessed by euglycemic hyperinsulinemic clamps. Fasting insulin may be lowered by fat.  MUFA is preferable to PUFA for fasting insulin and glucose lowering. The addition of 3-4g of N3 fats will lower TG and BP and reduce the proportion of subjects with metabolic syndrome. Dairy fat (50% saturated fat) is also related to a lower incidence of metabolic syndrome in cohort studies.

Reviewer 2 Report

The author presents interesting data on the role of dietary fats in the treatment of metabolic syndrome. The topic is of interest and the data is well exposed in terms of presence of confidence intervals. However, there are some  concerns I recommend addressing:

Abstract: the results could be explained in a clearer way

Methods should be more thoroughly explained as a paragraph on it is lacking completely.

The way the review is organized now is somewhat confusing, as chapter 1. is both an introduction and focuses on the lipids abnormalities seen in MS. I suggest making an actual introduction first.

The review would benefit from a conclusion paragraph.

There is currently no discussion on controversial data, what is the author opinion? Perhaps a different dietary pattern could be responsible for controversial data in absence of macronutrient proportion shifts?

Ketogenic diets are increasingly popular and are very different from low carb diets, and they could be both high fat and low-fat diets on top of being very low carb. I recommend discussing about this although metanalyses might be lacking.

Author Response

Response to reviewer 2

I have  added an introduction, aims, methods and conclusions. 

Abstract:  Background. Metabolic syndrome increases the risk of CVD over and above that related to type 2 diabetes. The optimal diet for the treatment of metabolic syndrome is not clear. Materials and Methods. A review of dietary interventions in volunteers with metabolic syndrome as well as studies examining the impact of dietary fat on the separate components of metabolic syndrome was undertaken using only recent meta-analyses if available. Results. Most of the data suggests that replacing carbohydrate with any fat, but particularly polyunsaturated fat, will lower TG, increase HDL cholesterol, lower blood pressure but have no effects on fasting glucose in normal volunteers or insulin sensitivity as assessed by euglycemic hyperinsulinemic clamps. Fasting insulin may be lowered by fat.  MUFA is preferable to PUFA for fasting insulin and glucose lowering. The addition of 3-4g of N3 fats will lower TG and BP and reduce the proportion of subjects with metabolic syndrome. Dairy fat (50% saturated fat) is also related to a lower incidence of metabolic syndrome in cohort studies.

Abstract:  Background. Metabolic syndrome increases the risk of CVD over and above that related to type 2 diabetes. The optimal diet for the treatment of metabolic syndrome is not clear. Materials and Methods. A review of dietary interventions in volunteers with metabolic syndrome as well as studies examining the impact of dietary fat on the separate components of metabolic syndrome was undertaken using only recent meta-analyses if available. Results. Most of the data suggests that replacing carbohydrate with any fat, but particularly polyunsaturated fat, will lower TG, increase HDL cholesterol, lower blood pressure but have no effects on fasting glucose in normal volunteers or insulin sensitivity as assessed by euglycemic hyperinsulinemic clamps. Fasting insulin may be lowered by fat.  MUFA is preferable to PUFA for fasting insulin and glucose lowering. The addition of 3-4g of N3 fats will lower TG and BP and reduce the proportion of subjects with metabolic syndrome. Dairy fat (50% saturated fat) is also related to a lower incidence of metabolic syndrome in cohort studies.

Reviewer 3 Report

This is a narrative review. Readers can no judge its usefulness if you don't put either aims or methods to retrieve the papers you are reviewing about, neither conclusions. 

Author Response

Response to reviewer 3

I have  added an introduction, aims, methods and conclusions. 

Abstract:  Background. Metabolic syndrome increases the risk of CVD over and above that related to type 2 diabetes. The optimal diet for the treatment of metabolic syndrome is not clear. Materials and Methods. A review of dietary interventions in volunteers with metabolic syndrome as well as studies examining the impact of dietary fat on the separate components of metabolic syndrome was undertaken using only recent meta-analyses if available. Results. Most of the data suggests that replacing carbohydrate with any fat, but particularly polyunsaturated fat, will lower TG, increase HDL cholesterol, lower blood pressure but have no effects on fasting glucose in normal volunteers or insulin sensitivity as assessed by euglycemic hyperinsulinemic clamps. Fasting insulin may be lowered by fat.  MUFA is preferable to PUFA for fasting insulin and glucose lowering. The addition of 3-4g of N3 fats will lower TG and BP and reduce the proportion of subjects with metabolic syndrome. Dairy fat (50% saturated fat) is also related to a lower incidence of metabolic syndrome in cohort studies.

Abstract:  Background. Metabolic syndrome increases the risk of CVD over and above that related to type 2 diabetes. The optimal diet for the treatment of metabolic syndrome is not clear. Materials and Methods. A review of dietary interventions in volunteers with metabolic syndrome as well as studies examining the impact of dietary fat on the separate components of metabolic syndrome was undertaken using only recent meta-analyses if available. Results. Most of the data suggests that replacing carbohydrate with any fat, but particularly polyunsaturated fat, will lower TG, increase HDL cholesterol, lower blood pressure but have no effects on fasting glucose in normal volunteers or insulin sensitivity as assessed by euglycemic hyperinsulinemic clamps. Fasting insulin may be lowered by fat.  MUFA is preferable to PUFA for fasting insulin and glucose lowering. The addition of 3-4g of N3 fats will lower TG and BP and reduce the proportion of subjects with metabolic syndrome. Dairy fat (50% saturated fat) is also related to a lower incidence of metabolic syndrome in cohort studies.

Round  2

Reviewer 1 Report

The paper has now much improved